# Perinatal Environmental Health Education Intervention to Reduce Exposure to Endocrine Disruptors: The PREVED Project

**DOI:** 10.3390/ijerph19010070

**Published:** 2021-12-22

**Authors:** Houria El Ouazzani, Simon Fortin, Nicolas Venisse, Antoine Dupuis, Steeve Rouillon, Guillaume Cambien, Anne-Sophie Gourgues, Pascale Pierre-Eugène, Sylvie Rabouan, Virginie Migeot, Marion Albouy-Llaty

**Affiliations:** 1Center of Clinical Investigation Inserm 1402, University Hospital of Poitiers, 2 Rue de la Milétrie, 86021 Poitiers, France; elhouria.sp@gmail.com (H.E.O.); simon.fortin.sp@gmail.com (S.F.); nicolas.venisse@chu-poitiers.fr (N.V.); antoine.dupuis@univ-poitiers.fr (A.D.); guillaume.cambien@univ-poitiers.fr (G.C.); pascale.pierre.eugene@univ-poitiers.fr (P.P.-E.); sylvie.rabouan@univ-poitiers.fr (S.R.); virginie.migeot@univ-poitiers.fr (V.M.); 2Faculty of Medicine and Pharmacy, University of Poitiers, 6 Rue de la Milétrie, 86031 Poitiers, France; 3BioSPharm Pole, University Hospital of Poitiers, 2 Rue de la Milétrie, 86021 Poitiers, France; anne-sophie.gourgues@chu-poitiers.fr; 4Ecology and Biology of Interaction, CNRS UMR 7267, 86073 Poitiers, France; 5APHP Laboratory of Pharmacology, GH Henri Mondor, 94010 Creteil, France; steeve.rouillon.c2i11@gmail.com

**Keywords:** environmental health promotion, pregnancy, endocrine disruptors, lifestyle change intervention, RE-AIM, bisphenol A, parabens

## Abstract

Environmental health promotion interventions may reduce endocrine disruptor (ED) exposure. The PREVED (PREgnancy, preVention, Endocrine Disruptors) project was developed to improve knowledge, to enhance risk perception, and to change exposure behavior. Our objective was to present the phases of the PREVED project using the RE-AIM method. PREVED intervention consisted of three workshops during pregnancy. Reach, adoption, and implementation phases were assessed with qualitative studies. Efficacy study consisted of a three-arm randomized controlled trial (RCT) on 268 pregnant women: (i) control group (leaflet), (ii) intervention group in neutral location, (iii) intervention group in contextualized location. The main outcome was the percentage evolution of participants who reported consuming canned food. Secondary outcomes were evolution of psycho-social scores, evolution of ED presence in urine, and ED presence in colostrum. The intervention adoption was centered on upper-privileged women, but implementation assessment showed that key features (highly practical intervention) seemed to be carried out and had initiated some behavior changes. A total of 268 pregnant women participated in the intervention and 230 in a randomized controlled trial (control group: 86 and intervention groups: 172). We found no significant differences in consumption of canned food and in percentage of women having a decrease of bisphenol A or parabens in urine, but we found a significant increase in the evolution of risk perception score and overall psychosocial score in intervention groups (respectively: +15.73 control versus +21.03 intervention, *p* = 0.003 and +12.39 versus +16.20, *p* = 0.02). We found a significant difference in percentage of women with butylparaben detection between control group and intervention groups (13% versus 3%, *p* = 0.03). PREVED intervention is the first intervention research dedicated to perinatal environmental health education in France. By sharing know-how/experience in a positive non-alarmist approach, it improved risk perception, which is key to behavior change, aiming to reduce perinatal ED exposure. Including women in precarious situations remains a major issue.

## 1. Background

Endocrine disruptors (EDs) are present everywhere in our daily life. Defined by the World Health Organization (WHO) as “exogenous substance or mixture that alters function(s) of the endocrine system and consequently causes adverse health effects in an intact organism, or its progeny, or (sub)populations”, they are widely distributed. Even at very low doses, EDs are likely to have endocrine-disrupting effects [1], and exposure to mixed EDs may have synergistic effects, for example, on fetal testes [2].

Pregnant women are exposed to many EDs such as Bisphenol A (BPA) and its chlorinated derivatives (ClxBPA) by dermal and oral route [3]. Indeed, BPA is found in plastics and water [4], ClxBPA are found in tap water due to water purification process using chlorine [5]. They are also exposed to parabens (PBs) through cosmetics and personal care products [6]. Only a few women change or intend to change their consumption habits during pregnancy [7]. The half-life of biomarkers is 6 h. It reflects exposure since the previous day.

Intrauterine exposure to environmental factors such as EDs is likely to have health consequences. As described in the Developmental Origins Hypothesis of Health and Diseases (DOHaD) theory, this exposure affects not only fetal development (nervous system disturbance, prematurity), but also the fetus’ future life (behavioral disturbances, early puberty) [1].

While late health promotion interventions have only a moderate impact, early interventions significantly reduce this risk [8]. Preventing exposure to environmental chemicals is a priority for reproductive health professionals and pregnant women [9]. Every mother-to-be should be particularly aware of exposure sources and their potential risks for her fetus; she should know how to minimize this exposure [10]. A few studies examining how to limit ED exposure used restrictive diet [11,12,13]. To our knowledge, no study has been aimed at reducing ED exposure by a health education program among pregnant women.

The purpose of the PREVED (PREgnancy preVention Endocrine Disruptors) project was to develop, implement, and evaluate a health education program focused on environmental health and ED exposure during pregnancy. The evaluation of this program used plural assessment tools, thereby facilitating understanding of the mechanism of action and intervention transferability.

## 2. Methods

### 2.1. PREVED Project

PREVED project is a population health intervention research (PHIR) on environmental health education for pregnant women carried out in France from 2015 to 2021, with three phases: development, implementation, and evaluation according to the “Reach, Efficacy, Adoption, Implementation, and Maintenance” (RE-AIM) method [14]. The entire PREVED methodology is described elsewhere [15,16].

### 2.2. RE-AIM Method

RE-AIM method proposes an intervention framework integrating macroscopic (political, environmental, and organizational components) and individual components [14].

#### 2.2.1. Reach, Adoption, and Implementation Assessment

Educational interventions are complex [17] because there are many components that interact with one another; their assessment process should include reach, adoption, and implementation.

Intervention depends on the behavior of both beneficiaries and facilitators, the number of people targeted, the number and variability of outcomes and its flexibility to be customized [18]. In 2019, Glasgow et al. recommended the RE-AIM evaluation model in which: “Reach” represents the percentage and characteristics of targeted people, “Effectiveness” refers to the assessment of positive and negative consequences, including behavior, quality of life, satisfaction of beneficiaries, and physiological judgement criteria, “Adoption” represents the proportions and representativeness of the intervention contexts, and “Implementation” represents the extent to which the intervention was delivered as expected. The actual evaluation by the beneficiaries themselves (effectiveness) is the result of “Efficacy * Implementation“. “Maintenance” means maintaining long-term behavioral change at the individual and community levels [14].

**Reach** in the PREVED project was defined by the number and diversity of pregnant women participating in the program. Their characteristics included age, gender, marital status, body mass index, educational level, place of birth, and precariousness EPICES score [19].

Program **adoption** was defined by a mixed approach: (i) quantitative with the number of offered workshops and the number of participants among eligible women; (ii) qualitative with an inductive methodological approach applied between January and April 2017, using the tensions characterizing the study; for example, the need to avoid alarmist approaches. A qualitative survey of ethnographic type was carried out as part of a master 2 in sociology, with immersion in the field of training, maintenance of a logbook, and data collection of 33 documents and writings, 5 participant observations, and 11 semi-directional interviews.

**Implementation** was assessed with a qualitative study carried out between November 2017 and February 2018 using different approaches: observations from 11 workshops (7 neutral and 4 contextualized), interviews with 3 facilitators and 4 pregnant women, and satisfaction assessment of 18 workshops representing the views of 111 participants. This study analyzed the key features of program delivery as adaptations during workshops to meet specific needs of participants, behavioral techniques to engage them and remove barriers. It clarified the way the program sought to make pregnant women change their lifestyle choices and how they perceived workshops.

**Maintenance** was defined as the extent to which programs had potential for sustainability. It was measured by the number of workshops done after the end of the study, the number of sites continuing to deliver the program and the efficacy outcomes evolution.

**Effectiveness** was measured by merging efficacy and implementation evaluations.

#### 2.2.2. Efficacy Assessment

##### Efficacy Study Design

To assess **efficacy**, an open-label monocentric, randomized controlled trial (RCT), parallel-designed was conducted. Eligible women were identified from the list of pregnancy declarations centralized by “*Protection Maternelle et Infantile*” or PMI (maternal and child protection). The modalities of recruitment were detailed in the protocol article [16]. Central random blind-generated allocation with 1:1:1 ratio was performed based on fixed blocks of three before t0. The allocation sequence was generated on Microsoft EXCEL^®^ (function RAND). Pregnant women were randomly assigned to one of the three groups in the 1st trimester of pregnancy: (i) Control group (leaflet on EDs); (ii) Intervention group in neutral location (leaflet on EDs and collective workshops in a meeting room); (iii) Intervention group in contextualized location (leaflet on EDs and collective workshops in the real-life pedagogical apartment).

Three workshops were conducted during the 2nd and 3rd trimester of pregnancy, they focused on three themes: (i) indoor air quality (animated by a medical advisor for indoor environments), (ii) nutrition (animated by dietician trained in environmental health), and (iii) personal care products (animated by a cosmetologist).

##### Efficacy Study Outcomes

The main outcome was percentage evolution of participants who reported consuming canned food before and after intervention. This percentage was determined through a consumption questionnaire (Q1) developed by our research team [3] and administered at t0, t + 2 months, and t + 14 months. Q1 explored the various food consumption that could be a source of exposure to EDs.

Secondary outcomes were mean score evolution of psychosocial dimensions such as risk perception, self-esteem, sense of coherence, locus of control. These scores were determined through a psychosocial questionnaire (Q2) also developed by our research team: the PREVED© questionnaire [15], and administered at t0, t + 2 months, and t + 14 months. Q2 was structured on the basis of the Health Belief Model (HBM) [20] and explores: (i) ED knowledge; (ii) Risk from EDs; (iii) Risk Assessment of EDs; (iv) Perceived Ability to avoid ED exposure. The efficacy of the program was therefore assessed in terms of the knowledge, attitudes, and practices (KAP) of participants towards ED exposure.

Finally, we assessed urinary presence or concentration of BPA, chlorinated derivatives of BPA (ClxBPA) monochloro, dichloro, trichloro, and tetrachloroBPA (MCBPA, DCBPA, TCBPA, and TTBPA, respectively), and PBs (methyl-, ethyl-, propyl-, and butyl-PB, and MePB, EtPB, PrPB, and BuPB, respectively) at t0, t + 2 months, at childbirth, and t + 14 months; and presence or concentration in colostrum of BPA, ClxBPA, MePB, EtPB, PrPB, and BuPB at childbirth.

BPA and ClxBPA in urine samples were assayed by Ultra High-Performance Liquid Chromatography coupled with tandem Mass Spectrometry (LC-MS/MS) according to the method developed and validated by our team [21]. The description of sampling, transport, and storage of samples before analysis is detailed in the protocol article [16] (Appendix A).

To compare groups, a first variable defined whether each sample was below the LoD (Limit of Detection), above the LoQ (Limit of Quantification), or between these two thresholds. A second dynamic variable between two urinary stages defined three categories of development: decreasing variable (e.g., drop from superior to LoQ to below LoD), stability of the variable or rising variable (e.g., from below LoD to an intermediate level between LoD and LoQ). Whenever possible, for left-censored data less than 80%, data imputation by minimum of LoQ divided by two (LoQ/2 for samples below the LoQ) and by Truncation k-Nearest Neighbor imputation (kNN-TN) was performed [22], enabling quantitative analysis.

### 2.3. Statistical Analysis

Intervention exposure was defined as “having participated in at least two workshops”. Sensibility analysis was defined being exposed, if having participated in at least one workshop.

A descriptive analysis was performed on sociodemographic data, questionnaire scores, and on presence or concentrations of EDs, with mean and standard deviation for quantitative variables, and percentage for qualitative variables; paired t-tests were used to compare mean differences and χ^2^-test was used to compare percentages.

The Q1 mainly quantifies the number of (a) canned tuna/week; (b) preserved sweetcorn/week; (c) other canned food/week; (d) total canned food consumption/week; (e) canned drinks/day; (f) plastic drink bottles/week. We compared the evolution from before to after intervention of these consumptions between control group (Group 1) and intervention groups (Group 2 + 3).

The Q2 mainly quantifies an overall psychosocial score summing four sub-scores: ED knowledge; Perceived Ability to avoid ED exposure; Risk from EDs (perceived severity score); Risk Assessment of EDs (perceived vulnerability score). A risk perception score was calculated based on perceived severity and vulnerability [15].

Main ED presence or concentration statistics were performed between control group (Group 1) and intervention groups (Group 2 + 3). Additional analysis was performed for sub-group analysis (Group 2 versus Group 3).

We performed statistical modelling with risk perception as dependent variable and EDs risk perception determinants [23], such as socio-economic status and age, as independent variables, with linear multiple regression. We performed the same statistical modelling with overall psychosocial score.

Per-protocol (PP) analysis was performed to assess the effect of workshops on consumption, psychosocial variables, and ED presence or concentrations. Depending on the conditions of use, χ^2^ or Fisher tests were carried out for percentage comparison. After imputation of ED concentration, Student or Wilcoxon tests were carried out for mean comparison. We used ANOVA (analysis of variance) to compare means of the groups.

The number of participants was calculated using a two-sided test (α = 0.05 and β = 0.20), according to EDDS (*Endocrine Disruptors Deux-Sèvres*) cohort results [3]. Our hypothesis was that intervention would decrease the percentage of women who consumed canned food by 23 points. Then, 58 participants were required for each group for a total of 174 pregnant women. We expected 20% lost to follow-up. *In fine*, 210 participants were required to be included in our study. Due to major colostrum loss, we included 63 supplementary participants. This amendment was approved by the Personal Protection Committee (protocol version 10 approved on 15 May 2018).

Statistical analyses were performed using SAS 9.4^®^ (Statistical Analysis Software 9.4, SAS Institute Inc, Cary, NC, USA), Microsoft EXCEL^®^ (2010, Bellevue, Washington, USA), and open-source software R (version 4.0.5 2020, R Core Team, Vienna, Austria).

## 3. Results

### 3.1. Reach, Adoption, and Implementation

#### 3.1.1. Reach

The absolute number of pregnant women participating in PREVED program was 268, but 230 women were included in the PREVED study. Their characteristics included age, gender, marital status, body mass index, educational level, country of birth, and EPICES score, which are presented in Table 1. Modifiable factors are presented in Table 2.

#### 3.1.2. Program Adoption

##### Quantitative Analysis

A total of 436 workshops were held during the period from May 2017 to August 2019: 287 in 2017, 103 in 2018, and 46 in 2019. A total of 232 (53%) workshops were held in contextualized location compared to 204 (47%) in neutral location (Table 3).

*In fine*, 135 (60 in neutral location, 75 in contextualized location) of pregnant women followed the workshop among 4393 eligible women who received informative postal mail with a prepaid envelope.

##### Qualitative Analysis

Our ultimate purpose being to reach all pregnant women including disadvantaged populations, we chose an underprivileged and multicultural neighborhood of Poitiers city (France) with families from low socioeconomic and educational status to implement the intervention. However, the ethnographic study noted exclusion of the most vulnerable populations. It appeared that the intervention had omitted a thorough diagnosis of its needs within a given territory.

#### 3.1.3. Implementation

The qualitative study showed that the workshops offered a friendly and dynamic atmosphere, which fostered exchanges. The adaptability of workshop facilitators was appreciated by the participants. Indeed, a great deal of information-sharing, and a lot of interaction occurred in a convivial atmosphere. The participants considered the contents highly practical and concrete. They said they had initiated some changes. The key features of program delivery seemed to be respected with adaptations made during delivery to meet the specific needs of participants. While the objectives of the workshops appeared clear and sufficient, they were perceived as too numerous. The contents in the contextualized place seemed clear and accessible, reassuring, and not guilt-inducing.

### 3.2. Efficacy: RCT Results

After randomization, three participants declined to participate. We excluded three other participants because of missing data. Most analyses were carried out on data from 230 participants: 78 randomized in control group (Group 1) and 152 in intervention groups (Group 2 + 3). Figure 1 shows the flow chart of PREVED study.

All participants were in a relationship; their average age was 33 years. A minority was in precarious situations (10.3% in Group 1 and 14.5% in Group 2 + 3). Table 1 presents socio-demographic data.

#### 3.2.1. Intent-to-Treat (ITT) Analysis

Table 4 presents the results of the Q1 analysis. Consumption in 2nd (first visit) and 3rd (second visit) trimester between Group 1 and Group 2 + 3 was different for preserved sweetcorn (*p* = 0.02) and for ready-made meals (24.7% versus 9.3%, *p* = 0.01). We found no significant difference in consumption of total canned food, canned tuna, other canned food, canned drinks, and plastic drink bottles. Table 5 presents the results of the Q2 analysis. We found no significant difference between Group 1 and Group 2 + 3 in risk perception score evolution or overall psychosocial score evolution. However, we noted a positive trend in both control and intervention groups. The subjective knowledge on EDs of Group.1 increased significantly more than Group 2 + 3.

Table 6 presents the results of the urine ED analysis between 2nd (first visit) and 3rd trimester (second visit). Out of 225 participants who underwent urine measurement, we did not find a significant difference in percentage of women having a decrease of BPA (26% control group versus 24% intervention groups, *p* = 0.94) or MePB presence (19% control group versus 32% intervention groups, *p* = 0.09). Appendix A presents the comparison results between neutral location (Group 2) and contextualized location (Group 3): no significant difference in ED levels was observed. Appendix A presents the comparison results of the colostrum analysis. Out of 147 pregnant women with colostrum measurement, we found a significant difference in percentage of women with BuPB detection between control group and intervention groups (13% versus 3%, *p* = 0.03).

After imputation by LoQ divides by two sets of censored data, we found no significant differences in mean concentrations of MePB in urine, either in BPA or MePB concentration in colostrum (Table 7). After imputation by kNN-TN, we found no significant differences in mean concentrations of MePB in urine and colostrum, but we found a statistical difference in BPA mean concentrations (Table 7).

#### 3.2.2. Per-Protocol-Analysis

The PP analysis found a significant increase in the evolution of both risk perception score and a significant difference between groups (+15.73 control group versus +21.03 intervention group, *p* = 0.003) and overall score (+12.39 control group versus +16.2 intervention group, *p* = 0.02) (Table 8). Both significant differences were confirmed by a multivariate analysis that included age, educational attainment, and maternal figure. Linear regression found a significant relationship only between intervention groups and scores (data not shown).

We compared mean scores between the three groups (Table 9), and we found the same results with an increase of the risk perception score among each group after intervention: +16.2 in Group 1, +20.3 in Group 2 (neutral location). and +21.4 in Group 3 (contextualized location) (*p* = 0.02), and an increase of the overall psychosocial score after intervention: +12.7 in Group 1, +16.0 in Group 2 and +16.4 in Group 3 (*p* = 0.03). We also found that fast-food consumption increased significantly after intervention among 32.7% of the participants of Group 1, versus 10.5% in Group 2 and 5.3% in Group 3 (*p* = 0.004).

In urine ED PP analysis (Appendix A), we did not find a significant difference in percentage of women having a decrease of BPA (23% versus 33%, *p* = 0.85) or PBs, e.g., MePB presence (20% Group 1 versus 33% Group 2 + 3, *p* = 0.09). In Colostrum ED PP analysis (Table 10), we found a statistical difference in percentage of women having BuPB detection (13% Group 1 versus 3% Group 2 + 3, *p* = 0.01).

In PP analysis defined by at least one workshop, urine ED analysis (Table 11), we found a significant difference in percentage of women having a decrease of MePB presence (18% Group 1 versus 33% Group 2 + 3, *p* = 0.04). In Colostrum ED analysis (Table 12), we found a statistical difference in percentage of women having BuPB detection (14% Group 1 versus 1% Group 2 + 3, *p* = 0.007).

### 3.3. Maintenance

The ITT analysis of consumption outcomes after one year from childbirth found no significant difference. However, we found in PP analysis that 42.7% of participants in intervention group, versus 39% in control group, significantly increased their consumption of canned tuna, and 46% had a stable consumption, versus 36% in control group (*p* = 0.04).

We found no significant difference for the intervention’s impact on the evolution of risk perception (*p* = 0.19 in ITT analysis and *p* = 0.58 in PP analysis). However, we found a significant effect of time. The risk perception score increased significantly during the follow-up time (*p* < 0.0001 in ITT and PP analysis) with mean scores of 62.25, 69.23, and 69.40 measured in the control group, respectively, in the second, third trimester of pregnancy, and one year after childbirth, versus 64.10, 69.50, and 70.31 in the intervention group in ITT analysis (66.45, 68.89, and 69.78 versus 61.74, 69.81, and 70.00 in analysis PP).

Out of 107 pregnant women who underwent urine measurement at one year after childbirth, we found a significant difference in percentage of women having a decrease of MePB presence (17% control group versus 44% intervention group, *p* = 0.02) in ITT analysis (Table 13) but not in PP analysis (Table 14).

## 4. Discussion

### 4.1. Main Results

PREVED is the first intervention research dedicated to perinatal environmental health education in France. The development, implementation, and evaluation of the PREVED project showed interesting results in terms of reach, adoption, and effectiveness.

#### 4.1.1. Reach and Adoption

We noted a lack of social diversity of participants. Despite the specific recruitment strategies through PMI department involved in DisProSe Consortium, the underprivileged population reached was limited.

The choice of collective workshops has many benefits. Peer education is an important health education instrument that facilitates development of new skills, acquisition of experiential knowledge and promotion of health behavior change through experience sharing and social support [24,25].

#### 4.1.2. Effectiveness = Efficacy ∗ Implementation

In the efficacy study, we found an effect on fast-food consumption, on risk perception score, and on MePB presence in urines. In the implementation evaluation, we found that the key features seemed to be carried out.

We did not find any significant effect on canned food consumption. However, the Q1 questions did not specify which kind of food containers. In fact, participants in workshops were encouraged to avoid metal food containers and to replace them by glass jars as much as possible, a point that was not clarified in the leaflet.

A more pronounced fast-food consumption effect was observed in the intervention groups with a major effect in contextualized location (Group 3).

The contextualized intervention was implemented in a pedagogical apartment, which was close to real-life. Home-based educational intervention on health could improve emotional care for chronic patients and their caregivers, especially for groups in precarious situations [26,27]. Likewise, a home-like environment could create added value in health promotion interventions [28,29].

However, we did not find major differences between Group 2 and Group 3. In fact, we noted that the apartment used for contextualized intervention was insufficiently exploited during workshops. The difference between the two intervention groups was consequently small, and our objective of promoting experimental knowledge was not fully attained.

The increased risk perception scores and a significant difference between control group and intervention groups were highlighted. This increase was maintained even more than a year from the intervention. In practice, risk perception represents a major lever and determinant of health behavior and motivation to change [30,31]; elevated risk perception could promote healthier and safer behaviors [32]. This may explain the reduction of the consumption of canned tuna observed in the long-term. Furthermore, both the interaction and sharing of experiences between participants are likely to contribute to favorable evolution. Involvement in health education processes depends largely on social factors such as peer support, especially within existing groups, and choice of a relatively familiar environment [33].

In PP and ITT analysis, a difference in BuPB presence in colostrum between control group and intervention groups was the only one highlighted (13% versus 1%, *p* = 0,03), as it was not significant for other EDs. In urine, we found no significant decrease between 2nd and 3rd trimesters. There were higher percentages of women having a decrease in MePB presence in the intervention groups, between control group and intervention groups (19% control group versus 32% intervention groups, *p* = 0.09). For BPA and ClxBPA, intervention failed to demonstrate a reduced detection level. We did not find that contextualization of workshops influenced biomarker levels in urine or colostrum. Urinary BPA detection percentage (43% 2nd trimester) was lower than in studies on pregnant women in France in 2011 (74%) [34] and in 2016 (100%) [35]. That said, among Canadian pregnant women, detected BPA in first trimester was 43% [36]. Concerning colostrum, in our study, BPA was detected at 77%, close to an American study on women who had just given birth [37]. However, comparison to the literature is complicated: detection percentage varies greatly (from 17 to 100%) due mainly to various analytical methods. ClxBPA, which is less studied than BPA, showed detection percentages close to those found in pregnant women of the EDDS cohort [3]. We detected more pronouncedly for 2nd trimester in PREVED study than in EDDS study: MCBPA (57% PREVED versus 34% EDDS), DCBPA (42% versus 34%), TCBPA (21% versus 35%), and TTBPA (31% versus 18%). In colostrum, because of the lipophilic nature and bioaccumulation of ClxBPA, we expected to find higher percentages of detection than those found. They were higher than those of EDDS (MCBPA 46% PREVED versus 18% EDDS, DCBPA 47% versus 23%, TCBPA 66% versus 17%, and TTBPA 51% versus 2%) [3].

Regarding PBs, proportions were lower than in other studies: a Japanese study [38], found stronger detection of MePB in pregnant women than our study (71% PREVED versus 94% Japanese study), EtPB (59% versus 81%), PrPB (22% versus 89%), and BuPB (3% versus 54%). Similarly, in an EDDS study on detection of biomarkers in urine at 2nd trimester of pregnancy in France, detection percentage was higher than in the PREVED study: MePB (71% PREVED versus 97% EDDS), EtPB (59% versus 77%), PrPB (22% versus 84%), and BuPB (3% versus 64%). First two PBs (MePB and EtPB) are most widely used in cosmetics, since many cosmetic products combine these ingredients so as to increase their antimicrobial potential [39]. In colostrum, we found the same detection gradient for PBs are comparably detected for 2nd trimester in PREVED and in EDDS: MePB (88% PREVED versus 90% EDDS), EtPB (60% versus 50%), PrPB (31% versus 30%), and BuPB (5% versus 27%). As mentioned above, a drop in PBs occurred at different times for MePB in urines, which was found in ITT and PP analysis.

Other RCTs aimed at reducing ED exposure through an intervention. A first study carried out in 2011 found a decrease in BPA detection percentages with a canned-food-excluded-diet in young adult population [11]. Another study, carried out in 2013, did not find any decrease in children and their parents’ urinary phthalates and BPA concentrations [13]. Two other studies carried out were aimed at reducing BPA and phthalate exposure in family members or young adult population, by avoiding plastic packaging and canned food. Both showed decreased mean concentrations for BPA in urine, by 66% for family members [12], and by 79% for young adults, respectively [40]. To our knowledge, only one RCT has been carried out in pregnant women in view of reducing phthalate exposure by eating only fresh and organic diet, without finding any decrease [41].

### 4.2. Strengths and Limits

#### 4.2.1. Evaluation Model

We adopted a parallel RCT with randomization at the individual level, with both quantitative and qualitative analyses. Our choices in this PHIR are questionable. Some authors have recommended in complex interventions to choose stepped-wedge cluster randomized trial where clusters are randomly allocated to different sequences, with each sequence defining the timing of cluster switch from the control to the experimental condition [42]. Even if individual randomization is rarely well-adapted to PHIR because interventions of interest are generally delivered at a group level (e.g., schools, health centers, geographical areas) [42], our targeted population (pregnant women) was not an identifiable group, and group contamination was consequently not possible.

Experimental methods for the evaluation of complex interventions, such as RCT, are the reference methods because they seek to determine the effects of the intervention with high internal validity, avoiding confusion bias. The RCT is clearly an option for PHIR [42], even if it has some drawbacks: (i) interventions are often carried out in the same controlled environment, excluding the interaction between intervention and environment from analysis; (ii) the recruited individuals are highly motivated, decreasing external validity [43]; (iii) the focus of the trial is on effectiveness (efficacy), which de facto excludes adaptation factors, targets achieved and institutionalization; (iv) RCTs are also limited for the study of behavioral factors [44] because the outcome is highly dependent on health determinants (individual characteristics, cultural and social environment, and health systems); (v) the standardization of the intervention, as expected in trials, is not favorable from a continuous learning perspective, which requires variation of the intervention [43].

Instead of RCT, some authors propose alternative assessment models such as the “out of control” test method, where only key functions are standardized while the form is adaptive [43] or RCT adaptations such as cluster randomized trials, pragmatic trials, cluster, and pragmatic or non-RCT designs (quasi-experimental, cohort, realistic evaluation, case-cohort studies) [17]. These new evaluation models raise questions of feasibility, acceptability, fairness, and sustainability of interventions and their adaptation to the results of qualitative studies, which provide better understanding of how and why we obtain these assessment results [44]. The process studies are largely designed to assess contextual factors and causative mechanisms [18].

We chose the RE-AIM model because it provided a solid framework to assess the implementation of the PREVED project. It enabled simultaneous examination of both participant-level outcome data and detailed organizational and site-level data [45].

#### 4.2.2. Outcome Choice of the Randomized Control Trial

The main outcome was canned food consumption and not biomarker presence or concentration. A pre/post comparison of biomarker presence could have been a relevant primary endpoint, as suggested by a California study in which a diet devoid of deleterious packages for three days led to a significant decrease in urinary concentrations of BPA metabolites from an average of 3.7 to 1.2 ng/mL [12]. However, other studies suggest that urinary BPA concentrations may vary during pregnancy [46,47,48], while urinary PB concentrations are minimally impacted [49]. These studies have not explored changed consumption patterns that may have occurred between two samples. As a result, it is difficult to conclude that changes in urinary concentrations are due solely to the physiology of pregnancy. That is why before/after comparison of biomarker presence in urine, was not the primary outcome in the PREVED study. Colostrum, which begins to form in the middle of pregnancy [50], may be a good biomarker of cumulative pregnancy exposure to lipophilic molecules such as BPA, Clx-BPA, and PBs. However, it is a rare and difficult matrix to collect under the conditions required to avoid BPA contamination (manual sampling, without gloves or breast pumps): only a single sample can be taken. Before/after comparison of biomarker presence in colostrum is impossible. We preferred to choose various outcomes: consumption, psychosocial, and biomarkers with an interdisciplinary contribution of analytical chemistry, social psychology, epidemiology, sociology, and health promotion disciplines.

#### 4.2.3. Construction

##### Consortium

The consortium creation involving the actors in the evaluation system from the construction of the intervention as part of a continuous improvement approach (researchers, field actors, decision-makers) has enabled scientific projects to mature, as recommended [51]. The solution science of the PHIR is strongly embodied in the practices and especially in the ability to support the meeting and sharing between different expertise to promote their hybridization and thereby the production of new expertise and a non-unidirectional transfer of knowledge [52]. However, in the PREVED project, this did not suffice insofar as there was a lack of exchange between partners; it could be called an epistemic misunderstanding [53]. As it is recommended that evaluation should be considered and built at the same time as the intervention itself, the consortium focused too early on the evaluation phase, which appeared too rigid to build an action of environmental health promotion.

##### Behavior Theory Model

The conception of PREVED workshops was based on the HBM including 12 behavior modification techniques from the 2013 taxonomy of Michie et al. [54]. According to the Medical Research Council, the main components of an intervention include the development of a theoretical model to better understand the change process [18]. Also, it has been interesting in the PREVED project to integrate sociocultural and economic factors [55], particularly social determinants of health and local context [56] using a solid theoretical base and a consistent and rigorous methodology [57].

##### Diagnosis with Pregnant Women

According to the PREVED sociologic study, the project omitted a thorough diagnosis of our target population of pregnant women’s needs. The behavior of pregnant women has been the subject of a normative and stereotypical approach. Despite the qualitative study on 12 women conducted before the cross-sectional study, a deeper exploration of lifestyle should have been performed. In citizen science, co-creation approach is recommended [58].

##### Public Heath Deployment

As part of evidence-based-health-promotion [59], this study could contribute to health policy deployment of the environmental health intervention in the care pathway during pregnancy. However, while this type of intervention has been disseminated in maternities, there is evidence of its impact. With the PREVED project, we have proposed effective intervention on risk perception and behavior change. To integrate these workshops into the pregnant women’s care pathway, it is necessary to involve medical doctors, nurses, and/or midwives. However, preventive medical recommendations of ED exposure avoidance during pregnancy are not yet being followed [60]. Medical doctors are not educating pregnant women on environmental risk prevention [7] because they are essentially focused on infectious biological risks. They also usually have non-specialized experimental knowledge of emergent risks. However, they have recently become increasingly cognizant of these risks [61].

We have previously recommended to educate pregnant women using simple words, taking the time to understand their representations, and questioning them about their knowledge, risk perception, and behavior (pre/post-test). This exchange could employ an educational tool, which could be part of the PREVED© questionnaire to help health professionals [15]. This new tool, which has been informatized in a smartphone via QR code available in general practitioners’ (GP) waiting rooms, has been tested and perceived by GPs as useful to initiate a discussion about environmental health with patients [62]

That said, we have described and reported on our public health interventions with the TIDieR tool, which like other tools (Astaire, Trend), does not make the distinction between key function (potentially transferable dimensions) and form (dimensions associated with translation with a specific context) proposed by Villeval, who introduced a “key function/implementation/context” model, which makes a distinction between transferable element assessment and context-specific element assessment [63]. This model is the best alternative for transferability insofar as it takes into account the context [64].

## 5. Conclusions

The PREVED intervention is the first intervention research dedicated to perinatal environmental health education in France. Results from program adoption at the institutional level, in terms of participant reach and effectiveness (efficacy ∗ implementation), suggest that the program could be widely implemented. However, it is necessary to co-construct the intervention along with a targeted population, which could consist in young women before pregnancy or more broadly, young people. Maintenance evaluation should contribute to the conservation of this intervention in the health pathways.

## Figures and Tables

**Figure 1 ijerph-19-00070-f001:**
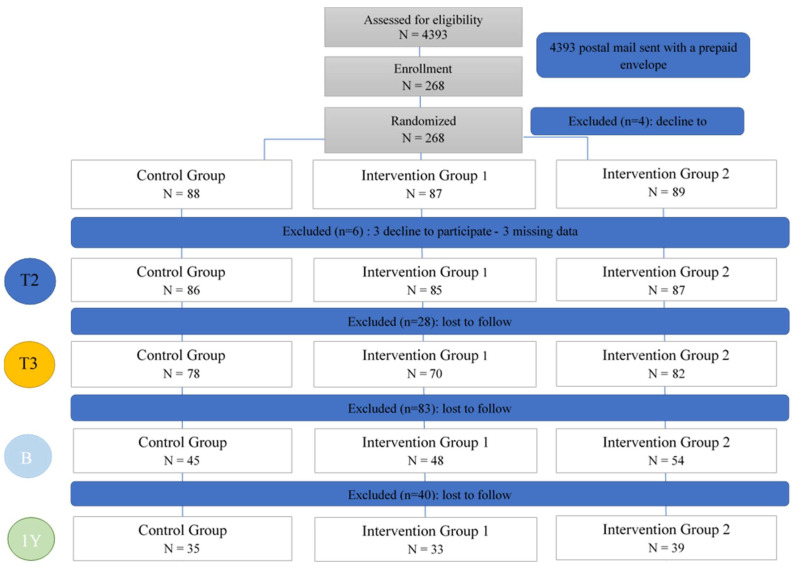
Flow Chart–PREVED Study.

**Table 1 ijerph-19-00070-t001:** Characteristics of PREVED cohort pregnant women.

	Control Group (n = 78)		Intervention Group (n = 152)	
	n1	%	n2	%
**Maternal age (years)**				
<30	14	17.9	29	19.1
30–34	35	44.9	73	40.0
≥35	29	37.2	50	32.9
Mean; ±SD	33.1	±4.3	32.8	±4.0
**Marital status**				
Married/Cohabiting/Civil partnership	78	100.0	152	100.0
**Parity**				
Nulliparous	19	24.4	50	32.9
1 child	25	32.1	60	39.5
≥2 children	34	43.6	42	27.6
**Women’s BMI (kg/m²)**				
<18.5	4	5.1	13	8.6
18.5–25	59	75.6	112	73.7
25–30	13	16.7	21	13.8
>30	2	2.6	6	3.9
Mean; ±SD	22.4	±3.2	22.6	±3.7
**Did you finish your studies?**				
No	11	14.1	8	5.3
Yes	67	85.9	141	92.7
**Women’s educational level**				
None	1	1.3	0	0.0
Primary, secondary school (<12th grade)	1	1.3	2	1.3
Certificate of Professional Competence	0	0.0	1	0.7
French baccalaureate (12th grade)	3	3.8	9	5.9
French baccalaureate +2 (12–14th grade)	10	12.8	22	14.5
Higher education (>14th grade)	59	75.6	112	73.7
Other	4	5.2	6	3.9
**Country of birth**				
Metropolitan France	70	89.7	144	94.7
Overseas France	2	2.6	0	0.0
Other	6	7.7	8	5.3
**EPICES score**				
Precarious situation (≥30.17)	8	10.3	22	14.5
No precarious situation (<30.17)	70	89.7	130	85.5
Mean; ±SD	8.3	±11.1	9.3	±12.2

**Table 2 ijerph-19-00070-t002:** Modifiable factors in the PREVED cohort pregnant women.

	Control Group (n = 78)		Temporal *p*-Value (First-Second Visit)	Intervention Group (n = 152)		Temporal *p*-Value (First-Second Visit)
	n1	%		n2	%	
**Smoking**						
*Three months before conception*						
No	64	82.1	0.26	113	74.3	**0.02**
Yes	14	17.9		39	25.7	
*During the first trimester*						
No	69	88.5		130	85.5	
Yes	9	11.5		22	14.5	
**Alcohol consumption ***						
*Before conception*						
No	4	5.1	**<0.001**	25	16.4	**<0.001**
Yes	74	94.9		127	83.6	
*Drinking more than 4 glasses during the 1st trimester * (n = 73 for control group; 147 for intervention group)*						
Never	69	94.5		124	84.4	
Very rarely	4	5.5		20	13.6	
Once a month	0	0.0		1	0.6	
Two or three times a month	0	0.0		2	1.4	
**Consumption of canned tuna (number/week)**						
*At the first visit*						
No	24	30.8	**<0.001**	43	28.3	**<0.001**
Yes	54	69.2		109	71.7	
Mean; ±SD * (n1 = 54; n2 = 108)	1.8	±1.7		1.8	±1.1	
*At the second visit*						
No	46	59.0		84	55.3	
Yes	32	41.0		68	44.7	
Mean; ±SD * (n1 = 32; n2 = 68)	1.5	±0.8		1.7	±1.0	
**Consumption of preserved sweetcorn (number/week)**						
*At the first visit*						
No	39	50.0	**0.002**	66	43.4	**0.003**
Yes	39	50.0		86	56.6	
Mean; ±SD * (n1 = 39; n2 = 86)	1.8	±1.9		2.2	±1.7	
*At the second visit*						
No	58	74.4		92	60.5	
Yes	20	25.6		60	39.5	
Mean; ±SD * (n1 = 21; n2 = 60)	1.3	±0.6		1.8	±1.4	
**Consumption of other canned food (number/week)**						
*At the first visit*						
No	18	23.1	**0.004**	28	18.4	**<0.001**
Yes	60	76.9		124	81.6	
Mean; ±SD * (n1 = 59; n2 = 123)	2.9	±2.9		2.3	±2.2	
*At the second visit*						
No	35	44.9		55	36.2	
Yes	43	55.1		97	63.8	
Mean; ±SD * (n1 = 43; n2 = 93)	2.2	±1.9		2.0	±1.4	
**Total canned food consumption (number/week)**						
*At the first visit*			**0.02**			
Mean; ±SD * (n1 = 71; n2 = 142)	4.7	±4.7		4.7	±3.8	**0.04**
*At the second visit*						
Mean; ±SD * (n1 = 59; n2 = 119)	2.9	±2.3		3.4	±2.4	
**Consumption of canned drink products (/day)**						
*At the first visit*						
No	66	84.6	0.97	124	81.6	0.37
Yes	12	15.4		28	18.4	
Mean; ±SD * (n1 = 13; n2 = 28)	1.7	±1.7		1.5	±1.3	
*At the second visit (n1 = 77; n2 = 151)*						
No	65	84.4		129	85.4	
Yes	12	15.6		22	14.6	
Mean; ±SD * (n1 = 12; n2 = 22)	1.4	±0.7		1.7	±1.4	
Consumption of consumption of plastic drink bottles (/week)						
*At the first visit*						
No	45	57.7	0.52	79	52.0	0.68
Yes	33	42.3		73	48.0	
Mean; ±SD * (n1 = 33; n2 = 72)	1.4	±1.1		1.4	±0.9	
*At the second visit * (n1 = 78; n2 = 151)*						
No	41	52.6		82	54.3	
Yes	37	47.4		69	54.7	
Mean; ±SD * (n1 = 37; n2 = 68)	1.4	±1.1		1.5	±1.1	
**Consumption of fresh fruit and vegetables (number/day)**						
Mean; ±SD	4.0	±1.5	0.48	3.8	±1.3	0.31
*At the second visit*						
Mean; ±SD	4.3	±1.7		4.1	±1.3	
**Consumption of organic fruit and vegetables**						
*At the first visit*						
No	14	17.9	0.67	33	21.7	0.57
Yes	64	82.1		119	78.3	
*At the second visit*						
No	12	15.4		29	19.1	
Yes	66	84.6		123	80.1	
**Consumption of fast-food (/month)**						
*At the first visit*						
No	38	48.7	0.63	70	46.1	0.08
Yes	40	51.3		82	53.9	
Mean; ±SD * (n1 = 40; n2 = 82)	1.6	±1.0		1.6	±0.9	
*At the second visit*						
No	35	44.9		55	36.2	
Yes	43	55.1		97	63.8	
Mean; ±SD * (n1 = 30; n2 = 73)	1.53	±0.8		1.6	±1.3	
**Consumption of ready-made meals (number/week)**						
*At the first visit*						
No	38	48.7	**<0.001**	70	46.1	**<0.001**
Yes	40	*51.3*		82	*53.9*	
Mean; ±SD * (n1 = 28; n2 = 39)	1.5	±0.9		1.5	±1.2	
*At the second visit*						
No * (n1 = 40; n2 = 91)	37	*92.5*		82	*90.1*	
Yes	3	*7.5*		9	*9.9*	
Mean; ±SD * (n1 = 21; n2 = 23)	1.5	±1.4		1.3	±0.8	

* Missing data; SD: standard deviation. Significant results are bolded.

**Table 3 ijerph-19-00070-t003:** Adoption of the workshops in the PREVED study.

	Workshop1 Indoor Air Quality	Workshop2 Food	Workshop3 Care Product	Total (n Workshops)
Total (n Workshops)	148	145	143	436
Neutral location	69	69	66	204
Contextualized location	79	76	77	232
2017	95	95	97	287
2018	36	35	32	103
2019	17	15	14	46

**Table 4 ijerph-19-00070-t004:** Consumption evolution. PREVED Study.

	Control Group (n = 78)		Intervention Group (n = 152)		*p*
	n1	%	n2	%	
**Consumption of canned tuna (number/week) * (N = 169)**					
Decrease	34	57.6	72	65.4	0.47
Stable	9	27.1	21	19.1	
Increase	16	15.3	17	15.5	
**Consumption of canned tuna (number/week)**					
No consumption in first visit and second visit	19	24.4	41	27.0	0.10
Consumption in first visit and no consumption in second visit	27	34.6	43	28.3	
Consumption in first visit and second visit	27	34.6	66	43.4	
No consumption in first visit and consumption in second visit	5	6.4	2	1.3	
**Consumption of preserved sweetcorn (number/week) * (N = 137)**					
Decrease	28	62.2	56	60.9	0.84
Stable	94	20.0	22	23.9	
Increase	8	17.8	14	15.2	
**Consumption of preserved sweetcorn (number/week)**					
No consumption in first visit and second visit	33	42.3	60	39.5	**0.02**
Consumption in first visit and no consumption in second visit	25	32.1	32	21.1	
Consumption in first visit and second visit	14	17.9	54	35.5	
No consumption in first visit and consumption in second visit	6	7.7	6	3.9	
**Consumption of other canned food (number/week) * (N = 198)**					
Decrease	38	56.7	66	50.4	0.69
Stable	15	22.4	35	26.7	
Increase	14	20.9	30	22.9	
**Consumption of other canned food (number/week)**					
No consumption in first visit and second visit	10	12.8	20	13.1	0.15
Consumption in first visit and no consumption in second visit	25	32.0	35	23.0	
Consumption in first visit and second visit	35	44.9	89	58.6	
No consumption in first visit and consumption in second visit	8	10.3	8	5.3	
**Total canned food consumption (number/week) * (N = 227)**					
Decrease	51	68.0	92	63.9	0.41
Stable	8	10.7	25	17.4	
Increase	16	21.3	27	18.7	
**Consumption of canned drink products (number/day) * (N = 55)**					
Decrease	7	43.7	21	53.8	0.66
Stable	2	12.5	6	15.4	
Increase	7	43.7	12	30.8	
**Consumption of canned drink products (number/day) * (N = 228)**					
No consumption in first visit and second visit	61	79.2	113	74.8	0.43
Consumption in first visit and no consumption in second visit	4	5.2	16	10.6	
Consumption in first visit and second visit	8	10.4	11	7.3	
No consumption in first visit and consumption in second visit	4	5.2	11	7.3	
**Consumption of consumption of plastic drink bottles (number/week) * (N = 136)**					
Decrease	11	23.9	27	30.0	0.76
Stable	20	43.5	36	40.0	
Increase	15	32.6	27	30.0	
**Consumption of consumption of plastic drink bottles (number/week) * (N = 229)**					
No consumption in first visit and second visit	32	41.0	60	39.7	0.71
Consumption in first visit and no consumption in second visit	9	11.5	22	14.6	
Consumption in first visit and second visit	24	30.8	51	33.8	
No consumption in first visit and consumption in second visit	13	16.7	18	11.9	
**Consumption of fresh fruit and vegetables (number/day)**					
Decrease	18	23.1	35	23.0	0.73
Stable	27	34.6	60	39.5	
Increase	33	42.3	57	37.5	
**Consumption of organic fruit and vegetables (number/day)**					
No consumption in first visit and second visit	7	9.0	21	13.8	0.75
Consumption in first visit and no consumption in second visit	5	6.4	8	5.3	
Consumption in first visit and second visit	59	75.6	111	73.0	
No consumption in first visit and consumption in second visit	7	9.0	12	7.9	
**Consumption of fast-food (number/month) * (N = 134)**					
Decrease	17	40.5	32	34.8	0.09
Stable	22	52.4	39	42.4	
Increase	3	7.1	21	22.8	
**Consumption of fast-food (number/month)**					0.7
No consumption in first visit and second visit	35	44.9	61	40.1	
Consumption in first visit and no consumption in second visit	12	15.4	19	12.5	
Consumption in first visit IT and second visit	28	35.9	63	41.4	
No consumption in first visit and consumption in second visit	3	3.9	9	5.9	
**Consumption of ready-made meals (number/week) * (N = 80)**					
Decrease	13	43.3	30	60.0	0.35
Stable	11	36.7	8	16.0	
Increase	6	20.0	12	24.0	
**Consumption of ready-made meals (number/week) * (N = 228)**					
No consumption in first visit and second visit	47	61.0	101	66.9	**0.01**
Consumption in first visit and no consumption in second visit	2	2.6	11	2.3	
Consumption in first visit and second visit	19	24.7	14	9.3	
No consumption in first visit and consumption in second visit	2	2.6	11	7.3	

* Missing data. Significant results are bolded.

**Table 5 ijerph-19-00070-t005:** Psychosocial variables (Intention-to-treat analysis) in PREVED Study.

	Control Group (n1 = 78)		Intervention Group (n2 = 152)		*p*
	Δm *	IC 95%	Δm *	IC 95%	
**1-Endocrine-disrupting chemicals knowledge (score/40.5)**	+3.55	[+2.68; +4.42]	+3.40	[+2.88; +3.90]	0.75
Sources of exposure to endocrine-disrupting chemicals (score/22)	+2.15	[+1.48; +2.82]	+1.97	[+1.54; +2.40]	0.64
Definition of endocrine disruptors (score/7)	+0.44	[+0.16; +0.71]	+0.36	[+0.20; +0.52]	0.63
Ability to name molecules (score/6.5)	+0.77	[+0.54; +1.00]	+0.83	[+0.66; +0.99]	0.69
Pathways of exposure to endocrine Disrupting chemicals (score/5)	+0.19	[+0.07; +0.31]	+0.23	[+0.15; +0.31]	0.59
**2-Perceived ability to avoid chemical exposure (/100)**	+7.09	[+2.33; +11.85]	+5.11	[+1.80; +8.42]	0.50
**3-Risk from endocrine disrupting chemicals (perceived severity score/100) to:**	+20.21	[+16.19; +24.23]	+19.20	[+15.77; +22.63]	0.72
The health of pregnant women	+7.16	[+1.40; +12.91]	+7.61	[+3.28; +11.94]	0.90
The newborn health	+8.09	[+2.28; +8.54]	+5.41	[+2.28; +8.54]	0.31
The adolescent health	+7.56	[+3.50; +11.63]	+4.73	[+2.05; +7.41]	0.24
The adult health	+6.06	[+1.38; +10.74]	+4.47	[+0.87; +8.08]	0.60
**4-Risk assessment of endocrine-disrupting chemicals:**					
Perceived vulnerability score/1400	+275.60	[+229.20; +322.00]	+256.40	[+228.60; +284.20]	0.46
Perceived vulnerability score/100	+19.69	[+16.37; +23.00]	+18.31	[+16.33; +20.30]	0.46
**Risk perception score (3+4)/100**	+19.50	[+17.12; +22.78]	+18.76	[+16.68; +20.84]	0.51
**Global score (1+2+3+4): Score/100**	+15.63	[+13.21; +18.05]	+14.45	[+13.00; +15.91]	0.38
**Subjective knowledge on endocrine Disrupting chemicals (score/100)**	+28.24	[+24.23; +32.26]	+19.77	[+16.51; +23.03]	**0.002**
**The information received on endocrine-disrupting chemicals were (score/100)**					
Understandable * (n = 74; 142)	−1.65	[−8.33; +5.03]	−1.35	[−6.14; +3.45]	0.99
Scientific * (n = 74; 142)	−2.39	[−7.39; +2.61]	+2.15	[−0.87; +5.18]	0.10
Realistic * (n = 74; 142)	+3.96	[−1.01; +8.93]	−0.36	[−3.85; +3.12]	0.15
Stressful * (n = 74; 142)	−1.91	[−6.05; +2.24]	−3.34	[−5.80; −0.88]	0.53
Complete * (n = 73; 142)	+6.30	[−0.18; +12.78]	+11.70	[+7.72; +15.68]	0.14
**Events during the pregnancy you are daily concerned with (score/100)**					
Pregnancy pain	−7.13	[−13.09; −1.17]	−0.93	[−5.07; +3.22]	0.09
The consequences of consumption of toxic substances for the child * (n = 77; 151)	+1.64	[−7.16; +10.43]	+1.18	[−4.61; +6.97]	0.93
Infectious diseases * (n = 76; 152)	−8.25	[−14.79; −1.71]	−6.80	[−11.53; −2.07]	0.72
Genetic diseases	−5.5	[−11.60; +0.50]	−5.48	[−9.63; −1.33]	0.98
Child illnesses linked to chemical exposure	−1.73	[−7.31; +3.85]	+0.22	[−3.06; +3.50]	0.52
**Concept of a healthy baby**					
Healthy birth weight* (n = 78; 151)	+5.21	[−0.63; +11.04]	−3.25	[−7.75; +1.25]	**0.03**
Full-term birth	+3.05	[−3.05; +9.15]	−1.47	[−5.51; +2.56]	0.21
Normal intelligence quotient (IQ)	+6.08	[−0.60; +12.75]	+6.19	[+1.76; +10.63]	0.98
Ability to have children	+7.50	[+1.18; +13.82]	+1.91	[−1.70; +5.51]	0.13
Normal puberty	+4.73	[+0.19; +9.27]	+1.64	[−2.15; +5.44]	0.33
Normal weight (no obesity; no overweight)	+1.24	[−2.80; +5.28]	+0.37	[−3.51; +4.24]	0.76
No asthma	+3.36	[−0.91; +7.63]	+0.96	[−3.06; +4.98]	0.42
No behavior disorders	+3.69	[−1.50; +8.89]	+4.64	−[+0.41; +8.87]	0.79
Can play like all the other children	−1.14	[−6.57; +4.28]	−0.21	[−4.47; +4.05]	0.80
Not get sick so often	+3.03	[−2.16; +8.22]	+0.05	[−4.18; +4.28]	0.40
Able to make friends and fit in	−0.99	[−6.32; +4.34]	+5.54	[+1.54; +9.53]	0.06
Successful professional life	+0.53	[−5.12; +6.17]	+5.96	[+1.94; +9.98]	0.12
Successful emotional life	−1.35	[−7.03; +4.33]	+5.32	[+1.59; 9.05]	**<0.05**
**Efforts towards avoiding chemical exposure**					
Financially	−2.30	[−7.25; +2.63]	+0.40	[−2.76; +3.57]	0.34
In terms of time * (n = 77; 152)	+6.36	[+0.98; +11.75]	+3.11	[−0.40; +6.62]	0.30
In terms of comfort * (n = 77; 152)	−1.00	[−8.12; +6.12]	−2.05	[−6.12; +2.03]	0.79
**Locus of control**					
Internal locus of control (score/6)	+0.02	[−0.03; +0.08]	−0.03	[−0.07; +0.01]	0.11
External locus of control: chance (score/3)	−0.01	[−0.10; +0.08]	+0.01	[−0.05; +0.06]	0.83
External locus of control: medical personnel (score/4)	+0.06	[−0.02; +0.14]	−0.03	[−0.08; +0.02]	0.06
**Sense of coherence**					
Comprehensive (score/5)	−0.06	[−0.24; +0.13]	+0.06	[−0.07; +0.19]	0.28
Meaningful (score/4)	−0.03	[−0.18; +0.13]	+0.02	[−0.11; +0.14]	0.67
Manageable (score/4)	−0.16	[−0.32; +0.01]	−0.09	[−0.20; +0.03]	0.46
**Rosenberg self-esteem scale (score/40)**	+0.56	[−0.05; +1.18]	+0.49	[+0.01; 0.96]	0.85
**Assessment of anxiety in general (score/100)**	−3.74	[−7,65; +0.16]	−1.05	[−3.66; +1.56]	0.25
**Anxiety evolution: before and after the questionnaire (score/100)**	−6.65	[−12.35; −0.96]	−4.76	[−8.93; −0.60]	0.60
**Important risk taking about**					
Professional life	−2.62	[−7.40; +2.17]	−1.63	[−5.19; +1.94]	0.75
Sport activities	+3.81	[+0.42; +7.20]	−0.09	[−3.02; +2.83]	0.11
Sexual practices	+1.26	[−1.34; +3.85]	+0.97	[−1.25; +3.20]	0.88
Road traffic	−0.81	[−3.84; +2.22]	+2.24	[−0.57; +5.05]	0.15
Use of substance	+0.82	[−2.10; +3.74]	−0.15	[−2.56; +2.26]	0.63

* Δm: mean difference between first and second visit. Significant results are bolded.

**Table 6 ijerph-19-00070-t006:** Urine biomarkers and exposition: Univariate analysis between 2nd (first visit) and 3rd trimester (second visit) (intent-to-treat analysis) in PREVED study.

	Control Group (n = 80)		Intervention Group (n = 145)		*p*
	N	%	N	%	
**Bisphenol A (BPA)**					
Rising indicator	25	31	47	32	
Same indicator	34	43	63	44	0.94
Decline indicator	21	26	35	24	
**BPA Mono-Chlorinated (MCBPA)**					
Rising indicator	15	19	34	23	
Same indicator	48	60	67	46	0.13
Decline indicator	17	21	44	31	
**BPA Di-Chlorinated (DCBPA)**					
Rising indicator	22	27	37	26	
Same indicator	39	49	75	52	0.91
Decline indicator	19	24	33	22	
**BPA Tri-Chlorinated (TCBPA)**					
Rising indicator	13	16	22	15	
Same indicator	55	69	99	68	0.94
Decline indicator	12	15	24	17	
**BPA Tetra-Chlorinated (TTBPA)**					
Rising indicator	14	17	22	15	
Same indicator	47	59	88	61	0.90
Decline indicator	19	24	35	24	
**MethylParaben (MePB)**					
Rising indicator	15	19	28	19	
Same indicator	50	62	71	49	0.08
Decline indicator	15	19	46	32	
**EthylParaben (EtPB)**					
Rising indicator	22	27	29	20	
Same indicator	35	44	59	41	0.22
Decline indicator	23	29	57	39	
**PropylParaben (PrPB)**					
Rising indicator	8	10	22	15	
Same indicator	62	78	99	68	0.33
Decline indicator	10	12	24	17	
**ButylParaben (BuPB)**					
Rising indicator	2	3	2	1	
Same indicator	77	96	139	96	0.64
Decline indicator	1	1	4	3	

**Table 7 ijerph-19-00070-t007:** Univariate analysis after data imputation (intent-to-treat analysis) in the PREVED study.

	Control Group (n = 93)		Intervention Group (n = 132)		*p*
	Mean (ng/mL)	*n*	Mean (ng/mL)	*n*	
LoQ divides by two (LoQ/2)					
**MethylParaben (MePB)**					
Second trimester (urine)	15.8	86	13.6	171	0.85
Third trimester (urine)	7	80	14.8	146	0.69
Birth (colostrum)	0.28	39	0.21	88	0.27
One Year (urine)	3	35	0.4	72	0.27
**Bisphenol A (BPA)**					
Birth (colostrum)	1.16	45	1	102	0.99
Truncation k-Nearest Neighbor (kNN-TN)					
**MethylParaben (MePB)**					
Second trimester (urine)	16	86	13.8	171	0.86
Third trimester (urine)	6.8	80	14.3	146	0.08
Birth (colostrum)	0.3	39	0.27	88	0.38
One year (urine)	1.4	35	0.4	72	0.29
**Bisphenol A (BPA)**					
Birth (colostrum)	1.53	45	1.21	102	**0.048**

Significant results are bolded.

**Table 8 ijerph-19-00070-t008:** Consumption and psychosocial variables (per-protocol analysis) in the PREVED study.

	Control Group (n1 = 81)		Intervention Group (n2 = 149)		*p*
	N	*%*	N	*%*	
**Consumption Questionnaire**					
**Consumption of canned tuna (number/week) * (N = 169)**					
Decrease	40	65.6	66	61.1	0.83
Stable	12	19.7	25	23.2	
Increase	9	14.7	26	15.7	
**Consumption of preserved sweetcorn (number/week) * (N = 137)**					
Decrease	27	60.0	57	62.0	0.94
Stable	11	24.4	20	21.7	
Increase	7	15.6	15	16.3	
**Consumption of other canned food (number/week) * (N = 198)**					
Decrease	41	58.6	63	49.2	0.44
Stable	15	21.4	35	27.4	
Increase	14	20.00	30	23.4	
**Total canned food consumption (number/week) * (N = 227)**					
Decrease	50	66.8	93	65.0	0.94
Stable	12	15.8	21	14.7	
Increase	14	18.4	29	20.3	
**Consumption of canned drink products (number/day) * (N = 55)**					
Decrease	13	61.9	15	44.1	0.05
Stable	0	0.0	8	23.5	
Increase	8	38.1	11	32.4	
**Consumption of consumption of plastic drink bottles (number/week) * (N = 136)**					
Decrease	15	30.6	23	26.4	0.85
Stable	20	40.8	36	41.4	
Increase	14	28.6	28	32.2	
**Consumption of fresh fruit and vegetables number (/day)**					
Decrease	21	25.9	32	21.5	0.67
Stable	31	38.3	56	37.6	
Increase	29	35.8	61	40.9	
**Consumption of fast-food (number/month) * (N = 134)**					
Decrease	16	33.3	33	38.4	0.12
Stable	19	39.6	42	48.8	
Increase	13	27.1	11	12.8	
**Consumption of ready-made meals (number/week) * (N = 80)**					
Decrease	15	57.7	28	51.8	0.45
Stable	4	15.4	15	27.8	
Increase	7	26.9	11	20.4	
**Psychosocial questionnaire**					
1-Endocrine-disrupting chemicals knowledge (score/40.5) **	+3.22	[+2.52; +3.93]	+3.57	[+2.99; +4.14]	0.47
2-Perceived ability to avoid chemical exposure (/100) **	+7.09	[+2.33; +11.85]	+5.11	[+1.80; +8.42]	0.50
3-Risk from endocrine-disrupting chemicals (severity score/100) **	+15.61	[+10.64; 20.58]	+21.69	[+18.67; +24.70]	**0.03**
4-Risk assessment of endocrine-disrupting chemicals (vulnerability score/1400) **	+222.00	[+187.00; +257.10]	+285.10	[+253.60; +316.6]	**0.008**
**Risk perception score (3+4)/100 ****	+15.73	[+12.96; +18.51]	+21.03	[+18.99; +23.07]	**0.003**
**Global Score (1+2+3+4)/100 ****	+12.39	[+10.57; +14.21]	+16.2	[+14.55; +17.83]	**0.002**

* Missing data ** Δm: mean difference CI: Confidence Interval. Significant results are bolded.

**Table 9 ijerph-19-00070-t009:** Consumption and psychosocial variables by groups (per protocol analysis) in the PREVED study.

	0 or 1 Workshop (n1 = 92)		2 or 3 Workshops in Neutral Location (n2 = 60)		2 or 3 Workshops in Contextualized Location (n3 = 75)		*p*
	N	%	N	%	N	%	
**Consumption Questionnaire**							
**Consumption of canned tuna (number/week) ***							
Decrease	42	64.6	29	65.9	33	57.9	0.90
Stable	14	21.5	8	18.2	14	24.6	
Increase	9	13.9	7	15.9	10	17.5	
**Consumption of preserved sweetcorn (number/week) ***							
Decrease	28	58.3	26	59.1	29	67.4	0.63
Stable	13	27.1	11	25.0	6	14.0	
Increase	7	14.6	7	15.9	8	18.6	
**Consumption of other canned food (number/week) ***							
Decrease	46	56.8	20	38.5	36	57.1	0.16
Stable	16	19.7	19	36.5	15	23.8	
Increase	19	23.5	13	25.0	12	19.1	
**Total canned food consumption (number/week) ***							
Decrease	55	63.2	37	63.8	48	67.6	0.60
Stable	13	14.9	12	29.7	8	11.3	
Increase	19	21.8	9	15.5	15	21.1	
**Consumption of canned drink products (number/day) ***							
Decrease	13	59.1	6	37.5	8	50.0	**0.03**
Stable	0	0.0	6	37.5	2	12.5	
Increase	9	40.9	4	25.0	6	37.5	
**Consumption of consumption of plastic drink bottles (number/week) ***							
Decrease	16	29.1	9	26.5	12	27.3	0.94
Stable	21	38.2	16	47.1	18	40.9	
Increase	18	32.7	9	26.5	14	31.8	
**Consumption of fresh fruit and vegetables (number/day)**							
Decrease	25	27.2	11	18.3	16	21.3	0.68
Stable	35	38.0	24	40.0	27	36.0	
Increase	32	34.8	25	41.7	32	42.7	
**Consumption of fastfood (number/month) ***							
Decrease	19	34.6	15	39.5	13	34.2	**0.004**
Stable	18	32.7	19	50.0	23	60.5	
Increase	18	32.7	4	10.5	2	5.3	
**Consumption of ready-made meals (number/week) ***							
Decrease	19	59.4	13	59.1	11	44.0	0.07
Stable	4	12.5	4	18.2	11	44.0	
Increase	9	28.1	5	22.7	3	12.0	
**Psychosocial questionnaire**							
**Risk perception score/100 ****	+16.18	[−24.23; +58.45]	+20.27	[−16.00; +48.37]	+21.41	[−9.15; +56.52]	**0.019**
**Global score/100 ****	+12.7	[−14.3; +42.6]	+16.0	[−6.9; +38.0]	+16.4	[−13.1; +45.5]	**0.026**

* Missing data ** Δm: mean difference CI: Confidence Interval. Significant results are bolded.

**Table 10 ijerph-19-00070-t010:** Colostrum biomarkers and exposition in univariate analysis, intervention group with at least two workshop (per protocol analysis) in the PREVED study.

	Control Group (n BPA = 56) (n Parabens = 48)		Intervention Group (n BPA = 91) (n Parabens = 79)		*p*
	N	%	N	%	
**Bisphenol A (BPA)**					
Superior to LoD	42	75	71	78	0.67
Inferior to LoD	14	25	20	22	
**BPA Mono-Chlorinated (MCBPA)**					
Superior to LoD	23	41	44	48	0.39
Inferior to LoD	33	59	47	52	
**BPA Di-Chlorinated (DCBPA)**					
Superior to LoD	26	46	43	47	0.92
Inferior to LoD	30	54	48	53	
**BPA Tri-Chlorinated (TCBPA)**					
Superior to LoD	39	70	56	62	0.32
Inferior to LoD	17	30	35	38	
**BPA Tetra-Chlorinated (TTBPA)**					
Superior to LoD	29	52	46	51	0.88
Inferior to LoD	27	48	45	49	
**MethylParaben (MePB)**					
Superior to LoD	44	92	68	86	0.34
Inferior to LoD	4	8	11	14	
**EthylParaben (EtPB)**					
Superior to LoD	31	65	45	57	0.40
Inferior to LoD	17	35	34	43	
**PropylParaben (PrPB)**					
Superior to LoD	14	29	25	32	0.77
Inferior to LoD	34	71	54	68	
**ButylParaben (BuPB)**					
Superior to LoD	6	13	1	1	**0.01**
Inferior to LoD	42	87	78	99	

LoD: Limit of Detection. Significant results are bolded.

**Table 11 ijerph-19-00070-t011:** Urine biomarkers and exposure n, univariate analysis between 2nd (first visit) and 3rd trimester (second visit), intervention group with at least one workshop (per protocol analysis) in the PREVED study.

	Control Group (n = 88)		Intervention Group (n = 137)		*p*
	n	%	n	%	
**Bisphenol A (BPA)**					
Rising indicator	28	32	44	32	
Same indicator	38	43	59	43	0.76
Decline indicator	22	25	34	25	
**BPA Mono-Chlorinated (MCBPA)**					
Rising indicator	16	18	33	24	
Same indicator	51	58	64	47	0.25
Decline indicator	21	24	40	29	
**BPA Di-Chlorinated (DCBPA)**					
Rising indicator	22	25	37	27	
Same indicator	44	50	70	51	0.85
Decline indicator	22	25	30	22	
**BPA Tri-Chlorinated (TCBPA)**					
Rising indicator	15	17	20	14	
Same indicator	60	68	94	69	0.84
Decline indicator	13	15	23	17	
**BPA Tetra-Chlorinated (TTBPA)**					
Rising indicator	15	17	21	15	
Same indicator	51	58	84	61	0.88
Decline indicator	22	25	32	24	
**MethylParaben (MePB)**					
Rising indicator	17	19	26	19	
Same indicator	55	63	66	48	**0.04**
Decline indicator	16	18	45	33	
**EthylParaben (EtPB)**					
Rising indicator	22	25	29	22	
Same indicator	40	45	54	39	0.32
Decline indicator	26	30	54	39	
**PropylParaben (PrPB)**					
Rising indicator	10	11	20	15	
Same indicator	69	79	92	67	0.16
Decline indicator	9	10	25	18	
**ButylParaben (BuPB)**					
Rising indicator	2	2	2	1	
Same indicator	85	97	131	96	0.66
Decline indicator	1	1	4	3	

Significant results are bolded.

**Table 12 ijerph-19-00070-t012:** Colostrum biomarkers and exposure in univariate analysis, intervention group with at least one workshop (per protocol analysis) PREVED study.

	Control Group (n BPA = 51) (n Parabens = 44)		Intervention Group (n BPA = 96) (n Parabens = 83)		*p*
	N	%	N	%	
**Bisphenol A (BPA)**					
Superior to LoD	39	76	74	77	0.93
Inferior to LoD	12	24	22	23	
**BPA Mono-Chlorinated (MCBPA)**					
Superior to LoD	21	41	46	48	0.43
Inferior to LoD	30	59	50	52	
**BPA Di-Chlorinated (DCBPA)**					
Superior to LoD	25	49	44	46	0.71
Inferior to LoD	26	51	52	54	
**BPA Tri-Chlorinated (TCBPA)**					
Superior to LoD	34	67	61	64	0.71
Inferior to LoD	17	33	35	36	
**BPA Tetra-Chlorinated (TTBPA)**					
Superior to LoD	25	49	50	52	0.72
Inferior to LoD	26	51	46	48	
**MethylParaben (MePB)**					
Superior to LoD	41	93	71	86	0.32
Inferior to LoD	3	7	12	14	
**EthylParaben (EtPB)**					
Superior to LoD	30	68	46	55	0.16
Inferior to LoD	14	32	37	45	
**PropylParaben (PrPB)**					
Superior to LoD	13	30	26	31	0.84
Inferior to LoD	31	70	57	69	
**ButylParaben (BuPB)**					
Superior to LoD	6	14	1	1	**0.007**
Inferior to LoD	38	86	82	99	

LoD: Limit of Detection. Significant results are bolded.

**Table 13 ijerph-19-00070-t013:** Urine biomarkers and exposure: Univariate analysis between 2nd trimester (first visit) and one year (intent-to-treat analysis) in the PREVED study.

	Control Group (n = 35)		Intervention Group (n = 72)		
	N	*%*	N	*%*	*p*
**Bisphenol A (BPA)**					
Rising indicator	11	32	14	19	
Same indicator	11	32	36	50	0.16
Decline indicator	13	36	22	31	
**BPA Mono-Chlorinated (MCBPA)**					
Rising indicator	5	14	11	15	
Same indicator	18	52	23	32	0.14
Decline indicator	12	34	38	53	
**BPA Di-Chlorinated (DCBPA)**					
Rising indicator	2	6	7	9	
Same indicator	21	60	40	56	0.77
Decline indicator	12	34	25	35	
**BPA Tri-Chlorinated (TCBPA)**					
Rising indicator	3	9	2	3	
Same indicator	25	71	56	78	0.40
Decline indicator	7	20	14	19	
**BPA Tetra-Chlorinated (TTBPA)**					
Rising indicator	9	26	9	13	
Same indicator	21	60	44	61	0.14
Decline indicator	5	14	19	26	
**MethylParaben (MePB)**					
Rising indicator	12	34	17	24	
Same indicator	17	49	23	32	**0.02**
Decline indicator	6	17	32	44	
**EthylParaben (EtPB)**					
Rising indicator	15	43	20	28	
Same indicator	11	31	37	51	0.14
Decline indicator	9	26	15	21	
**PropylParaben (PrPB)**					
Rising indicator	4	12	15	21	
Same indicator	25	71	47	65	0.48
Decline indicator	6	17	10	14	
**ButylParaben (BuPB)**					
Rising indicator	2	5	3	4	
Same indicator	32	92	66	92	0.89
Decline indicator	1	3	3	4	

Significant results are bolded.

**Table 14 ijerph-19-00070-t014:** Urine biomarkers and exposure: Univariate analysis between 2nd trimester (first visit) and one year, intervention group with at least two workshops (per protocol analysis) in the PREVED study.

	Control Group (n = 43)		Intervention Group (n = 64)		*p*
	N	*%*	N	*%*	
**Bisphenol A (BPA)**					
Rising indicator	12	28	13	20	
Same indicator	16	37	31	49	0.48
Decline indicator	15	35	20	31	
**BPA Mono-Chlorinated (MCBPA)**					
Rising indicator	6	14	10	16	
Same indicator	20	46	21	33	0.35
Decline indicator	17	40	33	51	
**BPA Di-Chlorinated (DCBPA)**					
Rising indicator	3	7	6	10	
Same indicator	25	58	36	56	0.91
Decline indicator	15	35	22	34	
**BPA Tri-Chlorinated (TCBPA)**					
Rising indicator	3	7	2	3	
Same indicator	32	74	49	77	0.65
Decline indicator	8	19	13	20	
**BPA Tetra-Chlorinated (TTBPA)**					
Rising indicator	9	21	9	14	
Same indicator	27	63	38	59	0.37
Decline indicator	7	16	17	27	
**MethylParaben (MePB)**					
Rising indicator	14	33	15	23	
Same indicator	19	44	21	33	0.09
Decline indicator	10	23	28	44	
**EthylParaben (EtPB)**					
Rising indicator	19	44	16	25	
Same indicator	13	30	35	55	**0.04**
Decline indicator	11	26	13	20	
**PropylParaben (PrPB)**					
Rising indicator	4	10	15	23	
Same indicator	32	74	40	63	0.17
Decline indicator	7	16	9	14	
**ButylParaben (BuPB)**					
Rising indicator	2	4	3	5	
Same indicator	40	94	58	90	0.82
Decline indicator	1	2	3	5	

Significant results are bolded.

## Data Availability

Dataset used for performing statistical analysis could be provided on reasonable request.

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
