# Peer review of "Perinatal Environmental Health Education Intervention to Reduce Exposure to Endocrine Disruptors: The PREVED Project"

_ijerph, 2021, doi:10.3390/ijerph19010070_

Round 1

Reviewer 1 Report

The investigators ran a three arm trial to assess the efficacy of workshops encouraging behavior changes to reduce endocrine disruptor exposure during pregnancy. Intervention workshops were compared to a leaflet control group with respect to risk perception, self-reported use of canned food, and urine biomarker levels.

The investigators conducted a through and well designed intervention with qualitative assessment of the implementation through interviews. The assessment of biomarkers is a major strength.

Introduction – add background on the duration reflected by the biomarkers assessed. Do they reflect meals yesterday, or patterns over a longer period.

L133 The investigators should clarify for the reader the intent of the meeting room vs. apartment as workshop locations and define ‘contextualized’ vs. ‘neutral’ earlier in the methods.

L141 Given that the intervention workshops covered three topics: air quality, nutrition, and personal care products, why was canned food considered the ‘main outcome’? Did they assess the other two realms?

L165 and Table 5 - It is not clear how the methods for assessing the temporal changes in levels of the compounds accounts for issues related to regression to the mean. If levels were extremely low or high at first visit, they are going to change at second look.  Also, if the levels were not high enough to be concerning to begin with, why are changes in these levels clinically important? Perhaps look at temporal changes in  absolute levels using regression modeling. Same with Table 9.

In Table 3, it seems like no consumption vs. consumption is coarse – perhaps modeling temporal changes in continuous number of meals would help tease out effects.

Table 1 – given that a number of the participants report smoking during pregnancy, was this issue addressed in the intervention?  Harmful effects are well documented – was this smoking rate assessed after workshops ?

Typo in Mean line of Total canned food consumption – n2

Table 1 – break into two tables:  Descriptive characteristics first. Then separate into another table with variables from “Smoking”  and below with some title like ‘Modifiable factors’.  Add statistics to this table – p-values. “Consumption of xx”  p-values should reflect change from first to second visit.

Add units for continuous variables – n of meals?

Clarify the difference between T2, T3 etc. and first /  second visit – use consistent terminology if the same.

Table 4 – clarify what mean difference refers to

Table 8 – typo , vs .

This is a robust manuscript with large amounts of interesting data in many tables. For readability, perhaps choose a subset of significant findings to keep in the main manuscript and move other null results to Supplemental tables to make the length more palatable.

L377 clarify direction of fast-food effect

L494 jumps from colostrum collection issues to urine?

Author Response

We would like to thank the reviewers for having appreciated our responses and for their new helpful comments. We addressed them in the following document and made the corresponding changes in the original manuscript, notably correction of grammatical errors by an American teacher. We believe that the new version is substantially improved by these inputs and we are grateful for the attention that was paid to our work.

Reviewer 1 comments:
The investigators ran a three arm trial to assess the efficacy of workshops encouraging behavior changes to reduce endocrine disruptor exposure during pregnancy. Intervention workshops were compared to a leaflet control group with respect to risk perception, self-reported use of canned food, and urine biomarker levels. The investigators conducted a through and well designed intervention with qualitative assessment of the implementation through interviews. The assessment of biomarkers is a major strength.

Introduction – add background on the duration reflected by the biomarkers assessed. Do they reflect meals yesterday, or patterns over a longer period.

We have added “ Half life of biomarkers is 6h. It reflects exposure since the last day”.

L133 The investigators should clarify for the reader the intent of the meeting room vs. apartment as workshop locations and define ‘contextualized’ vs. ‘neutral’ earlier in the methods.

The intent of the workshop location was to contextualize prevention message, as healing-oriented design of health care facilities is  known to reduce stress (Geimer-Flanders J Creating a healing environment: rational and research overview. Cleveland clinic journbal of medicine 2009; 76:S66.)

L141 Given that the intervention workshops covered three topics: air quality, nutrition, and personal care products, why was canned food considered the ‘main outcome’? Did they assess the other two realms?

The main outcome was canned food consumption because in 2014 only those data were  available. Biomarker analysis assesses  exposure  via food, air, and personal care products. Finally, other exposure  will be evaluated subsequently by specific variables

L165 and Table 5 - It is not clear how the methods for assessing the temporal changes in levels of the compounds accounts for issues related to regression to the mean. If levels were extremely low or high at first visit, they are going to change at second look.  Also, if the levels were not high enough to be concerning to begin with, why are changes in these levels clinically important? Perhaps look at temporal changes in absolute levels using regression modeling. Same with Table 9.

L165 : To compare groups, a first variable defined whether each sample was below the LoD (Limit of Detection), above the LoQ (Limit of Quantification) or between these two thresholds. A second dynamic variable between two urinary stages defined three categories of development: decreasing variable (e.g. drop from superior to LoQ to below LoD), stability of the variable or rising variable (e.g. from below LoD to an intermediate level between LoD and LoQ). Whenever possible, for left-censored data less than 80%, data imputation by minimum of LoQ divided by two (LoQ / 2 for samples below the LoQ) and by Truncation k-Nearest Neighbor imputation (kNN-TN) was performed [22], enabling quantitative analysis.

We totally agree with the reviewer, we are looking into the feasibility of a modeling which integrates temporal changes in both consumption and ED biomarkers for the three data-gathering phases: Second Trimester, Third Trimester and One year from childbirth.
We will probably use the latent class models to distinguish  subgroups of pregnant women and then to define different patterns of their consumption.
This modeling will be performed through collaboration with the biostatistics team of our  CIC-1402 lab and will be the subject of a future publication, in order to avoid overburdening the actual manuscript, which is already dense.

In Table 3, it seems like no consumption vs. consumption is coarse – perhaps modeling temporal changes in continuous number of meals would help tease out effects.

We wrote that “The Q1 mainly quantifies the number of a) canned tuna/week; b) preserved sweetcorn/week; c) other canned food/week; d) total canned food consumption/week; e) canned drinks/day; f) plastic drink bottles/week”. If number of canned tuna/week was null, then we decided to categorize into the no consumption group.

We have tried to express the variable in two ways to highlight the consumption evolution between two points, instead of modelling.

Table 1 – given that a number of the participants report smoking during pregnancy, was this issue addressed in the intervention?  Harmful effects are well documented – was this smoking rate assessed after workshops ?

Yes, workshops were about environmental health and so this issue was addressed in the intervention but we did  not assess  tobacco prevalence afterwards because there exist many interventions on the subject  during pregnancy so we could not  attribute a possible decrease to our intervention.

Typo in Mean line of Total canned food consumption – n2

We have changed it.

Table 1 – break into two tables:  Descriptive characteristics first. Then separate into another table with variables from “Smoking”  and below with some title like ‘Modifiable factors’.  Add statistics to this table – p-values. “Consumption of xx”  p-values should reflect change from first to second visit.

We broke the Table 1 into two tables and added in new Table 2 the temporal p values.

Add units for continuous variables – n of meals?

We added number/week in each meal consumption line.

Clarify the difference between T2, T3 etc. and first /  second visit – use consistent terminology if the same.

We replaced T2 by fist visit and T3 by second visit in Tables and  we added (first visit) and (second visit) in text after 2nd trimester and 3rd trimester respectively.

Table 4 – clarify what mean difference refers to

In the note, we added : * Δm : mean difference “between first and second visit”

Table 8 – typo , vs .

We replaced , by . Sorry for that.

This is a robust manuscript with large amounts of interesting data in many tables. For readability, perhaps choose a subset of significant findings to keep in the main manuscript and move other null results to Supplemental tables to make the length more palatable.

We put tables 7, 8 and 12 in supplementary files.

L377 clarify direction of fast-food effect

L314: We also found that fast-food consumption increased significantly after intervention among 32.7% of the participants of Group 1, versus 10.5% in Group 2 and 5.3% in Group 3 (p=0.004).

L377: In the efficacy study, we found an effect on fast-food consumption

L380: A more pronounced fast-food consumption effect was observed in the intervention groups with a major effect in contextualized location (Group 3).

L494 jumps from colostrum collection issues to urine?

We have changed the order of sentences to clarify our having not chosen urine biomarker or colostrum biomarker as primary outcome . L483

“The main outcome was canned food consumption and not biomarker presence or concentration. A pre/post comparison of biomarker presence could have been a relevant primary endpoint, as suggested by a California study in which a diet devoid of deleterious packages for three days led to a significant decrease in urinary concentrations of BPA metabolites from an average of 3.7 ng/mL to 1.2 ng/mL [12]. However, other studies suggest that urinary BPA concentrations may vary during pregnancy [46–48], while urinary PB concentrations are minimally impacted [49]. These studies have not explored changed consumption patterns that may have occurred between two samples. As a result, it is difficult to conclude that changes in urinary concentrations are due solely to the physiology of pregnancy. That is why before/after comparison of biomarker presence in urine, was not the primary outcome in the PREVED study. Colostrum, which begins to form in the middle of pregnancy [50], may be a good biomarker of cumulative pregnancy exposure to lipophilic molecules such as BPA, Clx-BPA and PBs. However, it is a rare and difficult matrix to collect under the conditions required to avoid BPA contamination (manual sampling, without gloves or breast pumps): only, a single sample can be taken. Before/after comparison of biomarker presence in colostrum is impossible. We preferred to choose various outcomes: consumption, psychosocial and biomarkers with an interdisciplinary contribution of analytical chemistry, social psychology, epidemiology, sociology, and health promotion disciplines”

Reviewer 2 Report

This manuscript presented the results of PREVED (PREgnancy, preVention, Endocrine Disruptors), the first intervention research dedicated to reducing ED exposure by perinatal environmental health education in France. Qualitative assessing studies showed that the intervention improved risk perception which is important to behavior change to reduce perinatal ED exposure. Base on research well designed and conducted, the manuscript is well written and provided very useful information in reducing ED exposure through perinatal health education. In my opinion this manuscript deserved publication in the journal of IJERPH after minor revisions addressing some deficiencies, as detailed below:

  1. The authors may have misunderstood the “Instructions for authors”. The words “Background”, “Methods”. ”Results” and ”Conclusion” should be deleted in the section of “Abstract”.
  2. Words like “best” at Line 72 are very objective and hardly justifiable. Generally, such words should be avoided in scientific writing, unless it can be backed by solid evidence.
  3. Line 90, please provide the year of the literature cited (Glasgow et al.). The same applied to “Michie et al.” at Line 513.
  4. The tables are lengthy and take much room, it may worth considering moving some table that are less to the point to supplementary information.
  5. While the reviewer agrees that this manuscript is structured fine, there seems too many sub-sections and the sub-section hierarchy is difficult to follow. The reviewer would recommend numbering the section and sub-section titles, and trimming some sub-sub-section titles when necessary.  

Author Response

We would like to thank the reviewers for having appreciated our responses and for their new helpful comments. We addressed them in the following document and made the corresponding changes in the original manuscript, notably correction of grammatical errors by an American teacher. We believe that the new version is substantially improved by these inputs and we are grateful for the attention that was paid to our work.

Reviewer 2 comments:
This manuscript presented the results of PREVED (PREgnancy, preVention, Endocrine Disruptors), the first intervention research dedicated to reducing ED exposure by perinatal environmental health education in France. Qualitative assessing studies showed that the intervention improved risk perception which is important to behavior change to reduce perinatal ED exposure. Base on research well designed and conducted, the manuscript is well written and provided very useful information in reducing ED exposure through perinatal health education. In my opinion this manuscript deserved publication in the journal of IJERPH after minor revisions addressing some deficiencies, as detailed below:

  1. The authors may have misunderstood the “Instructions for authors”. The words “Background”, “Methods”. ”Results” and ”Conclusion” should be deleted in the section of “Abstract”.

We deleted the subheadings.

  1. Words like “best” at Line 72 are very objective and hardly justifiable. Generally, such words should be avoided in scientific writing, unless it can be backed by solid evidence.

We have changed by “The evaluation of this program used plural assessment tools, thereby facilitating understanding of the mechanism of action and intervention transferability.”

  1. Line 90, please provide the year of the literature cited (Glasgow et al.). The same applied to “Michie et al.” at Line 513.

In 2019, Glasgow et al. recommend the RE-AIM evaluation model

The conception of PREVED workshops was based on the HBM including 12 behavior modification techniques from the 2013 taxonomy of Michie et al. [54].

  1. The tables are lengthy and take much room, it may worth considering moving some table that are less to the point to supplementary information.

We put tables 7, 8 and 12 in supplementary files.

  1. While the reviewer agrees that this manuscript is structured fine, there seems too many sub-sections and the sub-section hierarchy is difficult to follow. The reviewer would recommend numbering the section and sub-section titles, and trimming some sub-sub-section titles when necessary.  

We put in a section and subsection hierarchy.